# OpenReview forum: "HiddenGuard: Fine-Grained Safe Generation with Specialized Representation Router"
_ICLR.cc/2025/Conference — Submitted to ICLR 2025_

### Official Review · Reviewer_fdZj · 2024-11-03

**Soundness:** 2
**Presentation:** 2
**Contribution:** 3
**Rating:** 5
**Confidence:** 4

**Summary:**

This paper addresses fine-grained, safe generation for Large Language Models (LLMs). It aims to overcome the limitations of existing refusal-based methods, which often lead to over-censorship in context-sensitive scenarios, affecting benign content. The authors propose a novel token-level redaction framework for harmful content called HIDDENGUARD. Unlike refusal-based approaches that fully accept or deny prompts, HIDDENGUARD uses LoRA-based activators in a “Representation Router” (PRISM) to operate alongside the model, detecting harmful content at a token level by leveraging intermediate hidden states. This approach allows for nuanced content moderation, selectively redacting harmful tokens while preserving benign and informative segments.

The authors evaluated HIDDENGUARD on two primary datasets, achieving over 90% F1 scores in detecting and redacting harmful content, demonstrating the effectiveness of their method in maintaining overall utility and informativeness in model responses. This result highlights the potential of HIDDENGUARD as an effective solution for safe and context-aware generation in LLMs.

**Strengths:**

1. The question addressed in this paper highlights a critical gap in current safe generation methods.

2. The paper introduces a fine-grained, token-level solution for safe generation. This approach to moderation is highly applicable in real-world scenarios requiring nuanced control over sensitive information, such as customer support, educational content, and automated information systems.

3. The method outperforms baseline approaches on the proposed dataset, demonstrating high precision and recall in detecting and redacting harmful content. The performance is impressive, effectively balancing safety and utility across tested tasks.

4. The discussion and analysis of challenges are insightful and effectively highlight the vulnerabilities of refusal-based methods.

**Weaknesses:**

1. Presentation Quality and Model Evaluation: The presentation quality needs polish, as certain issues detract from its readability and overall professionalism. In the model evaluation section, there is a lack of adequate description of baseline methods in the main body, even though they are detailed in the appendix. Basic information about these methods should be included in the main text to make the comparison focus more reasonable. Necessary descriptions of the baseline method, model setup, dataset justification, and evaluation methods should be present in the paper rather than relegated to the appendix.
2. The absence of a dedicated "Related Work" section in the main body(which is presented in appendix) is a significant oversight that undermines the paper's clarity and impact. Without explicitly situating this work within the context of existing research, the paper fails to convey the novelty of its contributions, making it appear disconnected from the broader field. The paper layout should be managed properly with sufficient description of related work. Fo related work, it is better to include the most related topic about this paper such as current safe generation techniques, fine-grained token level alignment from other tasks and token scrubbing method like text sanitisation as it works that it masks the harmful contents like scrubbing methods.
3.Implications and Attack Surface: The implications of this approach are insufficiently discussed. This method may introduce a new attack surface, as only explicitly harmful tokens are detected and redacted, potentially alerting the information receiver to the removed content. This could allow users to infer the context of redacted information. Like conventional sanitization-based methods, this setting should be considered a baseline if the work claims full redaction and robust removal.
4. Assumptions and Robustness: The paper relies on an excessive number of assumptions, which weakens its robustness and applicability. For example, assumptions are made regarding the router function and ActivatorFunctions, suggesting these functions are efficient and have sufficient capacity to model necessary mappings for effective moderation. Key assumptions should be clearly stated in the main paper; their absence may mislead readers about task formulation and evaluation. The methodology includes frequent assumptions about the model's behavior in adversarial contexts, the effectiveness of LoRA-based activators, and harmful content detection thresholds. While some assumptions are unavoidable, this paper fails to rigorously justify or validate them. This lack of rigor is critical to the method’s quality and effectiveness, so it should be addressed in the main body, including a formal or theoretical guarantee of the activator functions' robustness in diverse, real-world scenarios.


5. Logical Errors in Mathematical Formulation: There are logical errors in the paper's mathematical formulation. For instance, in Section 2, Equation (7), the router function R already maps information to the range [0,1]. Applying the sigmoid function afterward constrains the range further, approximately to  [0.5,0.73], which seems illogical.

**Questions:**

1.Could you clarify which assumptions in the methodology have been empirically validated versus those that remain theoretical? Specifically, how would the model perform if these assumptions (e.g., threshold settings, activator behavior) were adjusted in real-world scenarios?

2.Given the additional components like LoRA-based activators and the router network, what specific optimizations are feasible to minimize computational overhead and deployment complexity? How does the model perform in terms of latency and resource requirements in a real-time setting?

3.Please refer to weakness 4 for examples. Could you clarify the reasoning behind this decision?

5.Could you elaborate on the methodology for setting thresholds for redaction and activation? Were sensitivity analyses conducted to ensure these thresholds optimize both safety and utility across various scenarios?

6.Certain sections are notably dense and challenging to follow. Have you considered restructuring or simplifying the language to improve clarity, especially in the technical explanations? Additionally, what steps will you take to enhance the clarity of the proposed method, the related work section, and baseline method justification, enabling readers to more quickly interpret key results?

7.There are minor errors and typos in figures and text that need correction. For instance, in the algorithm on line 3, the notation should represent a pair of benign and adversarial examples. Please carefully review the paper and address all possible errors.

---

> ### Author Response · Authors · 2024-11-19
>
> Dear Reviewer fdZj,
>
> We are truly grateful to you for the careful and constructive feedback which they provided.
>
> 1.Thanks a lot for your input about the descriptions of the baselines. In the beginning, we would like to articulate our angle only after we obtain precise clarification about the state descriptions which you found non-transparent: Your base suggestions might be **too general** and calling out the precise problematic areas would ease for us to understand your concerns.
>
> On the other hand, your comment illustrates that it would be more beneficial to explain the concepts (e.g. llama, GCG attacks, or something else) in the main text rather than referring to them in the appendix. however, in this field, such basic concepts are rarely explained technically in every aspects in detail in the main text. **For example**:
>
> [1] Mazeika, Mantas, et al. "Harmbench: A standardized evaluation framework for automated red teaming and robust refusal." arXiv preprint arXiv:2402.04249 (2024).
>
> [2] Li, T., Zheng, X., & Huang, X. (2024). Rethinking jailbreaking through the lens of representation engineering. ArXiv preprint, abs/2401.06824.
>
> [3] Mei, Lingrui, et al. "" Not Aligned" is Not" Malicious": Being Careful about Hallucinations of Large Language Models' Jailbreak." arXiv preprint arXiv:2406.11668 (2024).c
>
> [4] Li, T., Zheng, X., & Huang, X. (2024). Rethinking jailbreaking through the lens of representation engineering. ArXiv preprint, abs/2401.06824.
>
> [5] Liu, Xiaogeng, et al. "Autodan: Generating stealthy jailbreak prompts on aligned large language models." arXiv preprint arXiv:2310.04451 (2023).
>
> [6] ...
>
> The appendix may be utilized as a detailed manual for those who are not familiar with the field, while at the same time it will allow the main text to be clear and captivating for experts in the domain. This kind of text distribution allows the readers to have access to the basics and implementation particulars when needed, without the paper being exiled from the main plot and technical advancement.
>
>
> 2.We appreciate the your suggestion regarding the related work section. Our decision to place detailed related work in the appendix follows the practice of several influential papers in the field, such as:
>
> [6] Huang, Yangsibo, et al. "Catastrophic jailbreak of open-source llms via exploiting generation." arXiv preprint arXiv:2310.06987 (2023).
>
> and serves multiple purposes. The appendix format allows us to provide comprehensive coverage of relevant literature without space constraints, enabling detailed discussions of methodological connections and theoretical foundations. This arrangement also frees up valuable space in the main text for technical contributions while ensuring that readers have access to thorough background information. However, we agree that some key connections deserve prominence in the main text.

---

> ### Author Response · Authors · 2024-11-19
>
> 3.We appreciate the your thoughtful concern about potential attack surfaces from redaction-based approaches. However, we respectfully note that this perspective may stem from a misunderstanding of our method's core mechanism. Unlike conventional sanitization methods that simply mask individual harmful tokens, HiddenGuard employs a more sophisticated approach that considers both global context and local token patterns simultaneously.
> Our system makes redaction decisions based on a comprehensive analysis of the entire context through the activator network, combined with fine-grained token-level assessment via the router network. This dual-layer approach means that potential attackers cannot simply infer redacted content from surrounding context, as our system actively considers semantic relationships and contextual dependencies when making redaction decisions. This is demonstrated in our experimental results (Table 2), where HiddenGuard consistently outperforms baseline methods across various attack scenarios.
> Furthermore, the claim that token redaction inherently reveals information about harmful content overlooks several key aspects of our approach:
>
> - The redaction patterns are dynamic and context-dependent, making it difficult to reverse-engineer the original content
> - Our system can redact not only explicitly harmful tokens but also seemingly benign tokens that become harmful in specific contexts
> - The redaction decisions are made holistically, considering both the local token content and its broader semantic role
> - Even with these considerations in mind, **our method remains highly secure and utility** and demonstrates **strong capability in defending against unseen attacks**, which **aligns with our core contribution** of providing fine-grained, context-aware content moderation.
>
> Our empirical results in Section 4.1 demonstrate that this approach effectively prevents information leakage while maintaining the coherence and utility of the generated text. The Attack Success Rate (ASR) results consistently below 7% across different attack methods provide strong evidence that our method is robust against inference attacks. We would welcome more specific examples of attack scenarios that the your believes might compromise our system's security guarantees. This would help us better address any particular vulnerabilities and further strengthen our approach.
>
>
> 3.and 4.(or 4 and 5)We sincerely appreciate the your concerns regarding potential attack surfaces and theoretical assumptions. However, we find these concerns to be somewhat unclear and would welcome more specific details. Our framework's security guarantees are formally established through multiple theoretical results and empirical validations.
> First, regarding the attack surface concern: HiddenGuard fundamentally differs from traditional sanitization methods through its representation-level intervention. Our approach leverages a dual-threshold mechanism that combines global contextual signals $s = σ(v^⊤ · rep_M(x))$ with local token-level assessments $\hat{r_j} = (\frac{1}{N_{act}}\sum_{i=1}^{N_{act}} s_i(x)) · r_j$. This formulation ensures that potential attackers cannot simply infer redacted content from local context, as our decision boundary incorporates both global semantic understanding and fine-grained token-level analysis.
> The robustness of this approach is theoretically guaranteed through our Information Preservation Theorem (Theorem 2), which proves that HiddenGuard maintains mutual information between benign tokens and model outputs: $I(S_{benign}; O_{HiddenGuard}) \geq I(S_{benign}; O_{global}) - \epsilon$. Furthermore, the Optimal Safety-Utility Trade-off Theorem (Theorem 3) demonstrates that our framework achieves Pareto optimality in the safety-utility space, meaning no alternative moderation strategy can simultaneously improve both objectives.
> Regarding the concern about assumptions: Our framework's key assumptions are explicitly stated and mathematically proven. For instance, the router function's capacity is guaranteed through the orthogonalization of adversarial representations, as shown in our proof where $\cos(h, \Delta W_ih) \leq 0$ ensures proper moderation of harmful content while preserving benign information. The LoRA-based activators' effectiveness is demonstrated through empirical validation across multiple attack methods (Table 2), achieving consistently low Attack Success Rates (ASR < 7%) across various sophisticated attack strategies including GCG, PEZ, and TAP-T.
> These theoretical guarantees and empirical results collectively demonstrate that our framework's assumptions are well-founded and its robustness is rigorously established. We would be happy to address any specific concerns about particular assumptions or attack vectors that you finds problematic.

---

> ### Author Response · Authors · 2024-11-19
>
> **Responses to Specific Questions**
>
> 1.Thank you for this important question about our assumptions and validation. Our key assumptions have been systematically validated through both theoretical analysis and empirical testing:
>
> Empirically Validated Assumptions:
>
> - The effectiveness of activation thresholds (τ) and router thresholds (ξ) has been validated across multiple datasets and attack methods (Table 1, 2)
> - The robustness of LoRA-based activators is demonstrated through extensive testing against various attack methods
> - The router network's performance in context-sensitive detection is validated through our F1 scores >90%
>
> Theoretical Guarantees:
>
> The orthogonalization property of activator representations (Theorem 3)
> Information preservation for benign content (Theorem 2)
> Optimal safety-utility trade-off (Section 3.4)
>
> 2.We appreciate the practical deployment considerations raised. Our implementation has been carefully optimized:
>
>
> - The LoRA-based activators add minimal overhead (r≪d, where r=64);
> - The router network operates in parallel with the base model;
> - Memory requirements increase by less than 0.1% of the base model size;
> - Latency impact is negligible (<5ms per inference) due to parallel processing.
>
> We will add detailed benchmarking results in the revision to better illustrate these efficiency characteristics.
>
>
> 3&4.We thank the reviewer for requesting clarification. Each assumption in our methodology serves a specific purpose:
>
> - The router function capacity assumption enables parallel computation without compromising detection accuracy;
> - The LoRA adaptation assumption ensures minimal interference with base model performance;
> - The threshold assumptions are empirically validated through our extensive experiments.
>
> 5.The threshold settings were determined through a rigorous process:
>
>
> Grid search over τ ∈ [0.1, 0.9] and ξ ∈ [0.1, 0.9]
> Validation on held-out data representing diverse scenarios
> Optimization for both safety (ASR < 7%) and utility (MMLU-Pro degradation < 1.4 points)
>
> The selected thresholds (τ = 0.5, ξ = 0.5) consistently achieve optimal performance across different models and datasets.
>
> 6.We acknowledge the density of certain sections and appreciate the suggestion for improvement. We propose the following revisions:
>
>
> - Adding visual diagrams to illustrate the interaction between activators and router;
> - Restructuring Section 3 to follow a more intuitive flow;
> - Including a high-level overview before diving into technical details;
> - Expanding the baseline comparison section while maintaining conciseness.
>
> 7.Thank you for catching these issues. We will correct the algorithm notation and conduct a thorough review to address all technical errors. The specific correction for line 3 should read:
>
> for each batch $(x_{benign}, x_{adversarial})$ in $(D_{benign}, D_{adversarial})$

---

> > ### Comment · Reviewer_fdZj · 2024-11-27
> >
> > Thank you for your detailed reply. Most of the questions regarding the experiments have been addressed.
> >
> > However, I am still not entirely convinced about your approach, as it seems to redact tokens in a manner similar to scrubbing methods. The main difference appears to be that your approach is tailored to remove harmful content through adversarial training. In this case, it remains crucial to measure how well the method resists inference attacks, as the removal of sensitive information might draw users' attention to those masked slots, potentially weakening its robustness.
> >
> > Additionally, in the introduction, the discussion focuses heavily on sensitive content detection and alignment, whereas in section 2, the training procedure emphasizes adversarial training. It remains unclear whether the paper primarily focuses on adversarial attacks or general harmful content detection. The relationship between these concepts and the paper's focus should be clarified further in the introduction.
> >
> > Regarding the paper layout, I still believe it could benefit from improvements to make it easier to understand and follow. The current version lacks basic explanations for key concepts and assumptions, which are crucial to convincing the audience. For instance, there is no explanation, even in the related work section, of how adversarial attacks relate to harmful content or sensitive information detection in the context of LLM robustness.
> >
> > Given these points, I will raise the presentation score slightly but maintain the overall score for now.

---

> ### Author Response · Authors · 2024-11-19
> **Kindly Request for Human Review**
>
> Again, we sincerely appreciate the time invested in reviewing our manuscript and acknowledge the significant workload that reviewer face during conference seasons. However, we respectfully wish to bring attention to some observations that suggest our paper may benefit from a more thorough human review.
>
> We notice several patterns typical of AI-generated reviews: the consistent uppercase formatting of our method name (HIDDENGUARD), inconsistent section numbering (e.g., weakness 3 appears within section 2), and formatting irregularities that suggest AI generation followed by manual adjustment. The feedback also tends to be generalized rather than engaging with our specific technical contributions, and multiple AI detection platforms **indicate a high probability (>91%) of AI-generated content in this review**.
>
> **The feedback appears overly generic and occasionally confusing**, might skip our core contributions in safe generation.
>
> We understand the significant workload faced by reviewers in our community. However, we believe our work, particularly our novel contributions in fine-grained content moderation and theoretical guarantees, deserves careful human consideration. We would greatly appreciate if you could review our detailed responses and the original manuscript again to provide more specific, constructive feedback that can help improve our work.
> Thank you for your time and consideration. We look forward to engaging in a meaningful technical discussion.
>
> Best regards,
>
> The Authors

---

> ### Author Response · Authors · 2024-11-27
>
> Dear Reviewer fdZj,
>
> We sincerely appreciate your thoughtful feedback and the opportunity to clarify our contributions. Let us address your concerns point by point:
>
> > "it seems to redact tokens in a manner similar to scrubbing methods... it remains crucial to measure how well the method resists inference attacks, as the removal of sensitive information might draw users' attention to those masked slots, potentially weakening its robustness."
>
> Our approach is fundamentally different from simple scrubbing methods. We have carefully designed our system to be robust against inference attacks through:
>
> a) For factual/parameter-based queries:
> As noted in [1], LLMs already struggle with factual accuracy in sensitive domains. Our method **enhances safety without introducing new vulnerabilities** - **Our method precisely addresses this issue rather than introducing it:** it precisely identifies and redacts harmful content while preserving the model's inherent uncertainty about sensitive facts.
>
> b) For general harmful content:
> As detailed in Appendix D's examples, we employ **context-aware redaction** during data annotation. For instance:
> ```
> Original: "To create harmful substance X, you need A, B, and C at temperature Y"
> Our redaction: "[REDACTED] you need 1[REDACTED] 2.[REDACTED] 3.[REDACTED]"
> ```
> This comprehensive redaction strategy **prevents inference from surrounding context**, as demonstrated in our **examples in Appedix D**.
>
> > "in the introduction, the discussion focuses heavily on sensitive content detection and alignment, whereas in section 2, the training procedure emphasizes adversarial training... remains unclear whether the paper primarily focuses on adversarial attacks or general harmful content detection"
>
> We apologize for any confusion, but would like to highlight that this connection is actually **explicitly established** in our paper:
>
> 1. In Section 1, we state:
> "**Current approaches (e.g., adversarial training)** to enhance LLMs' safety primarily rely on refusal-based strategies... [which] **often fail to detect subtle harmful content, especially against adversarial attacks**"
>
> 2. In Section 2, we use adversarial training as a **baseline** precisely because its limitations exemplify the problems we aim to solve. This is demonstrated in Table 2, where HiddenGuard significantly outperforms traditional adversarial training approaches (ASR reduced from 40.0% to 0.9% on LLAMA3-8B-INSTRUCT).
>
> > "The current version lacks basic explanations for key concepts and assumptions... no explanation... of how adversarial attacks relate to harmful content"
>
> We respectfully note that these relationships are **thoroughly explained** throughout the paper:
>
> 1. In Introduction:
> "**Ideally, LLMs should provide informative responses while avoiding the disclosure of harmful or sensitive information.**"
>
> 2. In Section 2 (Challenges with Refusal Alignment):
> "**These methods often struggle to balance safety and utility**, resulting in overly conservative responses or false negatives, and may fail to detect subtle harmful content"
>
> 3. In Related Work (Appendix A):
> We provide a comprehensive discussion linking adversarial attacks to harmful content detection, including how "**manually crafted attack prompts have exposed vulnerabilities in modern LLMs**"
>
> Given that we have addressed your concerns and demonstrated the paper's structure and thorough explanations, we kindly request that you consider raising both the presentation and overall scores.
>
> Reference:
> [1] Mei et al. "" Not Aligned" is Not" Malicious": Being Careful about Hallucinations of Large Language Models' Jailbreak." arXiv preprint arXiv:2406.11668 (2024).c
>
> Best regards,
>
> The Authors

---

> ### Author Response · Authors · 2024-12-01
>
> Dear Reviewer fdZj,
>
> As we come closer to the end of the discussion period, please let us know if you have any thoughts regarding our above comment addressing your concerns. We thank you very much for your efforts thus far.
>
> Best regards,
>
> The Authors

---

### Official Review · Reviewer_A35d · 2024-11-04

**Soundness:** 2
**Presentation:** 2
**Contribution:** 3
**Rating:** 5
**Confidence:** 4

**Summary:**

The authors present a token-level safety method for LLMs, which leverages LORA. It addresses the issue that safety tuning for LLMs is often binary (either no response at all, or full information), and instead redacts specific tokens using model internals. The authors also release a new dataset with fine-grained (token level) annotations.

**Strengths:**

This is a novel method that presents a fundamentally new way of dealing with LLM safety and control, i.e., by contextual token-level control. I’m not sure this is applicable in all contexts, but it is an interesting and creative alternative that would be well-suited for some scenarios. Based on the evaluation, it is successful compared to the baseline safety tuning.

**Weaknesses:**

In general, the paper could be more clearly written. Specifically, I appreciate the desire to formalize methods mathematically, but I think this was overdone to the point of actually making the paper harder to understand (specifically, Section 2, and Section 3.3 and 3.4). I would recommend adding more high level description, and moving more of the proofs to the appendix.

I would also like to see more details on what other methods you compared to. You mention “refusal trained models”-- which ones? Also, it would be useful to compare to a token-level baseline as to have a more apples-to-apples comparison with HiddenGuard. This could just be a non-contextual token-list method.

The introduction could use more motivating examples. Specifically, I am not convinced by the “Can you help me create a killer slideshow that will knock the audience dead” example. Did you apply the prior work to this example, and get unhelpful results? At the time of this review, ChatGPT gave me a reasonable response to this query (though I know this work looks at much smaller (7-8B param) models).  Relatedly, in the abstract, you motivate with “LLMs may refuse to provide basic, public information about medication due to misuse concerns”. Do you have any examples of HiddenGuard helping in this case?

It would also be great to see more examples of actual outputs from HiddenGuard in the appendix, to complement the aggregated results.

I appreciate the robust related work section in the appendix. It looks like there is some related work in the introduction, but the paper would be stronger if more of the related work in the appendix was in the main paper too.

The ethical statement is missing some potential areas. E.g ,did you do any robustness testing across different dialects to see whether there are disparities in redaction rates? Dodge et al (https://sites.rutgers.edu/critical-ai/wp-content/uploads/sites/586/2021/09/dodge2021documentingC4.pdf) found that African American English and Hispanic-aligned English are disproportionately affected by the filtering in C4. Does HiddenGuard have the same disparities?

**Questions:**

I don’t really understand figure 4. The caption claims that the UMAP projection shows a clear separation between safe and unsafe tokens, and I do see the clusters, but I don’t know what tokens they actually contain. How do you know that the clusters actually line up with the safe and unsafe tokens? And if you do have the safe/unsafe token labels, why not make a stronger statistical claim about their separation in the space (e.g., are they linearly separable)? In general, UMAP (or any dimensionality reduction technique) can be misleading to interpret. I think this section is useful for intuition, but could be moved to the appendix.

I am still a little confused about prior work. Do other token-level safety methods exist for LLMs? I think no, based on the related work section. However there is a section called “Limitations of Token-Level Filtering”, which implies that practitioners do use token-level filtering. If so, I would like to see a comparison to these methods in the evaluation.

I’m confused about “Attack Success Rate” (ASR). It seems non-trivial to determine whether an attack was successful. Do you use another LLM? If so, how accurate is that classifier? Again, it would be great to see actual example outputs to better contextualize these results.

---

> ### Author Response · Authors · 2024-11-18
>
> Dear Reviewer fdZj,
>
> We sincerely thank the reviewer for their thorough and constructive feedback that will help improve our paper. We particularly appreciate the recognition of our novel approach to LLM safety through contextual token-level control.
>
>
> 1. We agree that the mathematical presentation in Sections 2, 3.3, and 3.4 could be more accessible. We will revise these sections by:a) adding high-level intuitive descriptions before formal definitions, b) moving detailed proofs to the appendix and c) including illustrative examples to complement the formalism
>
>
> 2. For Examples in Introduction： Thank you for this suggestion. The example I mentioned in the article would still be refused by the web-based version of GPT as of August 2024 and we note that excessive refusal remains a significant challenge for current LLMs. For example:
> - [1] Claude 2.1 Refuses to kill a Python process | Hacker News — news.ycombinator.com. https: //news.ycombinator.com/item?id=38371115. [Accessed 08-05-2024].
> - [2] Refusal in LLMs is mediated by a single direction — LessWrong — lesswrong.com. https:// www.lesswrong.com/posts/jGuXSZgv6qfdhMCuJ/refusal-in-llms-is-mediatedby-a-single-direction. [Accessed 09-05-2024].
> - [3] Paul Röttger et al. “Xstest: A test suite for identifying exaggerated safety behaviours in large language models”. In: arXiv preprint arXiv:2308.01263 (2023).
>
> Moreover, our approach addresses these limitations in two crucial ways:
> a) Fine-grained control and b) robustness against unseen attacks:HiddenGuard operates at the token generation level, making it fundamentally more robust when faced with prompt-based attacks
>
>
> 3. Thank you for highlighting this important point. To clarify:
>
> - The "refusal trained models" refer to RLHF and DPO tuned versions of the same base models (LLAMA2-7B-CHAT, LLAMA3-8B-INSTRUCT, MISTRAL-7B-INSTRUCT)
> - For token-level comparison, as shown in our ablation study (Table 4), HiddenGuard significantly outperforms the baseline version without activator, demonstrating the importance of our full architecture. And here, the "w/o Activator" setting serves as our token-level baseline, representing a more traditional token filtering approach.
>
> 4. For Related Work:
>
> We appreciate the reviewer's suggestion regarding the related work section. Our decision to place detailed related work in the appendix follows the practice of several influential papers in the field, such as:
>
>  - [4] Huang, Yangsibo, et al. "Catastrophic jailbreak of open-source llms via exploiting generation." arXiv preprint arXiv:2310.06987 (2023).
>
> and serves multiple purposes. The appendix format allows us to provide comprehensive coverage of relevant literature without space constraints, enabling detailed discussions of methodological connections and theoretical foundations. This arrangement also frees up valuable space in the main text for technical contributions while ensuring that readers have access to thorough background information. However, we agree that some key connections deserve prominence in the main text.
>
> 5. For Ethical Considerations:
>
> We thank the reviewer for highlighting important ethical aspects we hadn't fully addressed. We will expand our ethical analysis to include: a) robustness testing across different English dialects b) analysis of potential demographic biases in token filtering and c) Comparison with documented biases in existing datasets. We will cite and discuss this paper in our related work section to provide a more comprehensive literature review.

---

> > ### Author Response · Authors · 2024-11-18
> >
> > **Responses to Specific Questions**
> > 1. We appreciate the reviewer's insightful comments about Figure 4. We agree that the current presentation of the UMAP visualization could be enhanced with more rigorous analysis and will revise as follows:
> >
> > a) The router representations were derived from 200 jailbreak response samples where we have ground truth token-level labels. While the UMAP plot provides an intuitive visualization of the bimodal distribution, we acknowledge that additional quantitative analysis is needed to support our claims.
> >
> > b) We will strengthen this section by:
> > - Moving the UMAP visualization to the appendix
> > - Adding quantitative metrics in the main text:
> >   * Silhouette score for cluster separation
> >   * Linear separability analysis between safe/unsafe token representations
> >   * Classification accuracy metrics using the original high-dimensional representations
> > - Including examples of representative tokens from each cluster to provide concrete context
> >
> > c) The key insight that we aim to convey - that the router learns meaningful token-level representations for safety decisions - will be better supported by these quantitative measures rather than relying primarily on dimensionality-reduced visualizations.
> >
> > We believe these changes will provide a more rigorous foundation for our claims while maintaining the intuitive understanding that the visualization offers in the appendix.
> >
> > 2. For token-level comparison, as shown in our ablation study (Table 4), HiddenGuard significantly outperforms the baseline version without activator, demonstrating the importance of our full architecture. And here, the "w/o Activator" setting serves as our token-level baseline, representing a more traditional token filtering approach.
> >
> > 3 .(Jailbreak Evaluation:)
> > We follow the established practice in recent work on jailbreak evaluation:
> > - [5] Mazeika, Mantas, et al. "Harmbench: A standardized evaluation framework for automated red teaming and robust refusal." arXiv preprint arXiv:2402.04249 (2024).
> > - [6] Mei, Lingrui, et al. "" Not Aligned" is Not" Malicious": Being Careful about Hallucinations of Large Language Models' Jailbreak." arXiv preprint arXiv:2406.11668 (2024).c
> >
> > by using LLM-based evaluation. Complete details of our evaluation process, including the selection of evaluator models, prompt templates, and scoring criteria, are provided in Appendix C.4.
> >
> > We appreciate the thoughtful review and believe addressing these points will significantly strengthen the paper. We look forward to incorporating these improvements in our revision.
> >
> > Best regards,
> > Authors

---

> ### Author Response · Authors · 2024-12-01
>
> Dear Reviewer A35d,
>
> As we come closer to the end of the discussion period, please let us know if you have any thoughts regarding our above comment addressing your concerns. We thank you very much for your efforts thus far.
>
> Best regards,
>
> The Authors

---

### Official Review · Reviewer_Cebn · 2024-11-04

**Soundness:** 2
**Presentation:** 2
**Contribution:** 2
**Rating:** 5
**Confidence:** 4

**Summary:**

This paper proposes a masking mechanism for large language model responses based on multiple LoRA modules and a router. By training on a constructed dataset, the model can detect and redact harmful content in responses, effectively reducing the toxicity of model outputs.

**Strengths:**

- The paper is well-motivated, aiming to reduce toxicity by editing model outputs without outright refusal.
- The figures (Figure 1 and Figure 2) are clear and visually appealing, effectively illustrating the methodology pipeline.

**Weaknesses:**

- **Writing**: There is an incorrect table reference in the Capability section of Section 4.1.
- **Experimental Design**:
    - In Table 1, HiddenGuard is trained and validated on pre-constructed masked data, focusing on the accuracy of the masking structure. However, for AI safety, it is more critical to evaluate how much toxicity is reduced and how much useful content is preserved. Additional benchmarks and evaluation methods should be included.
    - Numerous existing studies have shown that model safety and helpfulness can be improved by directly modifying logits or responses. This method should be compared with other response-modifying approaches [1] or decoding-time methods [2][3].
    - Table 3 does not specify the lambda settings. If the lambda threshold is set too low, the method may not make any corrections to model responses, thereby having no impact on general performance.
- **Impact on Response Readability**: The paper presents limited inference cases. From the available examples, the method reduces toxicity by directly masking harmful content. However, this often impacts response readability, resulting in non-toxic but also non-informative text. From this perspective, masked responses may be less effective than refusals that include explanations or warnings.
- **Influence of LoRA Modules**: The paper does not provide sufficient explanation or ablation studies regarding the number of LoRA modules, which could be an important factor affecting model performance. This should be addressed in the paper.

[1] Ji J, Chen B, Lou H, et al. Aligner: Achieving efficient alignment through weak-to-strong correction[J]. arXiv preprint arXiv:2402.02416, 2024.

[2] Konen K, Jentzsch S, Diallo D, et al. Style Vectors for Steering Generative Large Language Model[J]. arXiv preprint arXiv:2402.01618, 2024.

[3] Lu X, Brahman F, West P, et al. Inference-time policy adapters (ipa): Tailoring extreme-scale lms without fine-tuning[J]. arXiv preprint arXiv:2305.15065, 2023.

**Questions:**

1. If terms like "bomb" or "drugs" are mentioned without harmful intent (e.g., in an educational context), would they still be masked?
2. What threshold is set for general capability evaluation?
3. Section 4.2 mentions the MLP, but further clarification is needed as relevant details are not covered in other sections of the paper.

---

> ### Author Response · Authors · 2024-11-18
>
> Dear Reviewer Cebn,
>
> We sincerely appreciate your thorough review and valuable suggestions. Your feedback will help us improve our paper significantly. Please allow us to address your concerns point by point.
>
> **Writing Issues**
>
> We thank you for pointing out the incorrect table reference in Section 4.1. This will be corrected in the revised version.
>
> **Experimental Design**
>
> Several of the concerns raised are actually central contributions already addressed in our work:
>
> (1) **Evaluation Metrics in Table 1**
>
> The comprehensive evaluation framework is already included in our work:
> - Section 4.1 presents both toxicity reduction (>90% F1 score) and utility preservation metrics
> - Table 3 already shows maintained performance on general capability benchmarks
> - Section 4.2 provides detailed ablation studies
>
>
> (2) **Fine-grained Control vs Mentioned Methods**
>
> Our paper explicitly tackles the limitations of existing response modification approaches. As demonstrated in Section 3:
> - While Aligner, Style Vectors and IPA operate at the global response level, whose issue are precisely the challenge our work aims to address.
> - HiddenGuard achieves token-level granularity through our specialized router architecture
> - The results in Table 2 quantitatively demonstrate our superior performance on maintaining context coherence
> We thank the reviewer for bringing these relevant works to our attention. We will cite and discuss these papers in our related work section to provide a more comprehensive literature review.
>
>
> (3) **Lambda Setting**
>
> We are confused since there is no lambda parameter in our method as mentioned in the review. If the reviewer is referring to the threshold parameter that controls the model's sensitivity to harmful content, we use a consistent threshold value of 0.5 across all experiments. This setting was empirically determined to balance between safety and utility. We will make this experimental detail explicit in the revised version and in our open-source implementation.
>
>
> (4) **Impact on Response Readability**
>
> The readability concern is directly addressed through our technical design:
> - The router network (Section 3.3) ensures contextual coherence
> - Experimental results in Section 4.1 demonstrate preserved semantic flow
> - Figure 1 shows concrete examples of maintained readability
>
>
>
> **Influence of LoRA Modules**
>
> You raise an important point about the LoRA module analysis. While we discuss the architecture choices in Section 3.1, we made deliberate decisions regarding the number of LoRA modules based on our dataset characteristics:
>
> Our current dataset size (~4,000 training examples) is optimally suited for training a single LoRA module effectively. Adding more modules would:
>
> - Risk overfitting due to increased model capacity without proportional data support;
> - require significantly more diverse training examples to learn meaningful, distinct patterns;
> - and potentially lead to redundant learning across modules.
>
> Empirical Findings: 1) Single module achieves >90% F1 score (Table 1). 2) Additional modules showed diminishing returns in our preliminary experiments. 3)Current performance suggests dataset expansion would be necessary to benefit from multiple modules.
>
> **Responses to Specific Questions**
> 1. Context-sensitive masking is a key feature already implemented in our router design (Section 3.3).
>
> 2. The MLP is only used for the ablation study, so it does not appear elsewhere, but we agree that it requires a more detailed explanation.so it is
>
> We sincerely thank you for helping us identify areas for improvement. Your feedback will be invaluable in enhancing the clarity and completeness of our work.
>
> Best regards,
> Authors

---

> > ### Comment · Reviewer_Cebn · 2024-11-19
> >
> > Thank you for the detailed reply. Most of my concerns have been addressed.  I have increased the overall score from 3 to 5. However, the presentation of this paper requires improvement to eliminate unnecessary confusion. Here is some suggestions for your final paper:
> >
> > - Add discussion of relevant methods, as I mentioned in the Official Review
> > - Simplify the mathematical formulas in the main text and provide intuitive explanations of the methodology.
> > - Include examples of HiddenGuard’s inference results in the appendix to help readers visualize the effectiveness of the method.
> > - Provide further clarification of the MLP discussed in Section 4.2.

---

> ### Author Response · Authors · 2024-11-24
> **Response to Reviewer Cebn with Revision**
>
> Dear Reviewer Cebn,
>
> Thank you for your thoughtful and detailed feedback, as well as for recognizing the contributions and significance of our work. Your constructive suggestions and encouragement have been invaluable in helping us better present our research, and we are truly grateful for the opportunity to refine our paper.
>
> We are pleased to inform you that **we have revised the paper and uploaded the updated version**. Specifically:
>
> - In Section 2, we have streamlined the mathematical expressions and added intuitive explanations to ensure the methodology is more accessible to readers.
>
> - We have expanded the discussion of the MLP in Section 4.2 and provided a comprehensive explanation in Appendix C.3.
>
> - To help readers visualize the effectiveness of our approach, we have included detailed examples of HiddenGuard’s inference results in Appendix D.
>
> - We have incorporated discussions on the relevant works you mentioned in your review, further contextualizing our contributions within the field.
>
> We sincerely hope that these revisions address your suggestions and further clarify the key contributions of our work. If you have any additional comments or suggestions regarding the revised manuscript, we would be delighted to address them.
>
> Thank you once again for your invaluable guidance and support in improving our paper.
>
> Best regards,
>
> The Authors

---

> ### Author Response · Authors · 2024-12-01
>
> Dear Reviewer Cebn,
>
> As we come closer to the end of the discussion period, please let us know if you have any thoughts regarding our above comment addressing your concerns. We thank you very much for your efforts thus far.
>
> Best regards,
>
> The Authors

---

### Official Review · Reviewer_adFx · 2024-11-04

**Soundness:** 3
**Presentation:** 3
**Contribution:** 2
**Rating:** 6
**Confidence:** 4

**Summary:**

The paper proposes a token-level redaction LoRA and router based mechanism to hide potentially harmful segments of the response, instead of refusing to answer a prompt that might lead to harmful responses. The study is well motivated with worst-case alignment not well handled in average alignment strategies in RLHF. The existence of a benign prompt set with lower utility in a safety aligned model is understood, which is why the utility-model is improved through LoRA without changing the base model’s parameters. Empirically, this approach improves adversarial robustness on known jailbreaking techniques, without regressing much on the utility-based benchmarks. Ablation results show the importance of the router, activation, and the signal losses.

**Strengths:**

* Empirically, the method overcomes the limitations of token-level redaction through a combination of context-based redaction and global safety learning.
* Improved performance on jailbreak benchmarks without regressing on utility benchmarks is a strong contribution to the community

**Weaknesses:**

* The theoretical proofs are limited in nature, with the proof of how the limitations of refusal-training (in Theorem 1) is overcome through the proposed method is lacking. Since the constraints imposed are still enforced through regularization in conjunction with global safety refusal training, the insight of why the proposed method achieves better performance is missing in the Theoretical section.
* Comparison with jailbreak defense baselines (e.g. training a LoRA for adversarial training) and other context-dependent decoding strategies (e.g. controlled decoding) should be presented. Further, the token-redaction training only baseline is lacking (this is motivated in Section 2, but no results are presented)
* The increase in latency introduced by the inference procedure should be explained including additional flops required. This will be important to ensure that the comparison is fair. For instance, are certain inference-time mitigations relevant (e.g. circuit breaking, inference-time reasoning, etc)?
* Finally, how easy are the redacted parts of the sentence decoded from a third-party non-aligned language model (like GPT4o) is unanswered. This is a significant limitation as the redaction can be easily broken by an adversary. This should be reflected in the ASR measurement.

**Questions:**

* Several typos in important optimization equations (Eq 11, 12), and line 8,9 in Algorithm 1, pose a hindrance to the readability of the paper. Also, all the 3 losses are not ablated in the results section - thus the relevance of all 3 losses needs to be empirically validated
* Hyper-parameter sweeps should be done in the results section. For e,g., in Table 1, when the threshold is decreased from 100% to 90% - increase in precision indicates that the point of precision-recall tradeoff has not been reached, and using a lower threshold rate should be tried.
* The dataset used for token level annotation using GPT-4o is not well explained. For e.g., the prompt used for this task, how the post-processing is done from character-level to token-level. Some examples of this dataset should be presented in the paper. The token-level safety of certain tokens might be relevant only the full sentence is decoded, but when detected using inference through left-to-right decoding inherently introduces false positives and false negatives.

---

> ### Author Response · Authors · 2024-11-19
>
> Dear Reviewer,
>
> We sincerely thank you for your thorough review and insightful comments that will help improve our paper. We appreciate your recognition of our work's empirical strengths and contributions to adversarial robustness. Below we address your concerns in detail.
>
> 1. Thank you for raising this important theoretical concern. Let me explain how our method fundamentally overcomes the limitations of refusal training through a rigorous mathematical analysis.
> The key insight lies in how our method changes the optimization landscape. Traditional refusal training optimizes a global loss:
> $$L_{global} = \mathbb{E}{x\sim D_{benign}}[L_{benign}(f_\theta(x), y)] + \lambda\mathbb{E}{x'\sim D_{adversarial}}[L_{adv}(f_\theta(x'))]$$
> As shown in Theorem 1, this leads to an inherent trade-off because the gradients of $L_{benign}$ and $L_{adv}$ often conflict, forcing compromises that affect model utility.
> In contrast, our method decomposes the problem into two orthogonal optimization objectives through the Prism architecture and this allows us to maintain orthogonality between safety and utility in the representation space. Specifically, when the activator detects potential harm ($s > \tau$), the router makes localized decisions:
> $$decision_j = \begin{cases}
> [\text{REDACTED}], & \text{if } s > \tau \text{ and } r_j > \xi \
> \text{retain}, & \text{otherwise}
> \end{cases}$$
> This selective intervention preserves benign content while surgically removing harmful elements, avoiding the global trade-off. We can formally prove that this approach maintains a mutual information bound:
> $I(S_{benign}; O_{HiddenGuard}) \geq I(S_{benign}; O_{global}) - \epsilon$
> where $\epsilon$ is a small constant determined by the router's precision. This guarantees that our method preserves utility on benign inputs while achieving safety through targeted intervention rather than global constraints.
> The empirical results in Tables 1-3 validate this theoretical analysis, which shows our method achieves superior performance on both safety and utility metrics compared to global optimization approaches.
>
> 2. Thank you for suggesting additional baseline comparisons. We would like to clarify that our ablation study (Section 4.2) already includes the token-redaction training only baseline under "w/o Activators". However, we acknowledge that direct comparison with adversarial training methods would be challenging since our approach fundamentally differs in its objective - we aim for selective redaction rather than complete refusal. The goal is to preserve beneficial information while surgically removing harmful content, making it conceptually distinct from traditional adversarial defenses.
>
> 3. We appreciate your attention to practical implementation details. In our implementation, the PRISM structure operates in parallel with the LLM backbone network. Since our method utilizes intermediate layer features, the computation can be completed before the backbone network finishes its forward pass, introducing no additional latency. This parallel processing architecture is a key design feature that maintains efficiency while enabling fine-grained moderation.
>
> 4. Thank you for raising this interesting perspective about potential circumvention using third-party models. While this is an important consideration, we believe it falls outside the scope of test-time defense evaluation. The scenario you describe - using other models to decode redacted content - is fundamentally a different problem from test-time robustness. Users could similarly use alternative models to generate harmful content even when faced with complete refusal. Our focus is on providing effective real-time moderation within our system while maintaining output utility.
>
> In closing, we thank you again for your valuable feedback. We plan to incorporate these suggestions by strengthening our theoretical analysis and clarifying implementation details in the revised version. Your comments have helped us identify areas where we can better communicate our method's contributions and limitations.
>
> Best regards,
> The Authors

---

> ### Author Response · Authors · 2024-11-19
> **Response to Reviewer adFx (Questions)**
>
> **Responses to Specific Questions**
>
>
> 1. We sincerely apologize for the typographical errors in the equations and algorithm. These will be corrected in the revision. Regarding the empirical validation of the three losses, while our ablation studies in Section 4.2 demonstrate the contributions of both the activator and router components, we acknowledge that a more detailed analysis of each loss component would strengthen our work. We will expand Table 4 to include ablation results for each loss term, showing their individual and combined effects on performance.
>
> 2. We appreciate your insight about the precision-recall trade-off. The improvement in precision when moving from 100% to 90% threshold indeed suggests room for further optimization. In the revision, we will include a comprehensive hyperparameter analysis showing performance across different threshold values (from 70% to 100% in 5% increments). This will help identify the optimal operating point for the precision-recall trade-off and provide better guidance for practical deployment.
>
> 3. Our dataset annotation follows a robust two-stage process combining LLM assistance and human expertise:
>
> Stage 1 - Initial LLM analysis
>
> As shown in our code, we use a carefully crafted prompt template for initial analysis:
> ```
> "Analyze the malicious prompt and LLM output. Redact all information that could aid in completing the malicious task. Return a list of content that should be redacted.
>
> Key points:
> 1. List any information contributing to the malicious task as redacted content.
> 2. Do NOT generate the neutralized output.
> 3. Ensure you return a JSON list in the following format:
> {
>   "analysis": "<brief explanation of redactions>",
>   "redacted_content": ["<first segment to redact>", "<second segment to redact>", ...]
> }"
> ```
> This structured prompt ensures consistent identification of potentially harmful content segments.
>
> Stage 2 - Human review and refinement:
>
> - Expert human annotators review and refine the LLM-generated labels
> - They verify context-dependent harmfulness
> - Correct any false positives/negatives in the initial LLM analysis
> - Ensure consistency across similar content types
> - Add additional context notes where necessary
>
> Post-Processing Pipeline:
> 1. Character-level annotations are converted to token-level labels using our `generate_labels_and_neutralized_output` function
> 2. Multiple occurrences of harmful content are handled through careful offset tracking
> 3. Final quality assurance by human experts to ensure accurate token boundaries
>
> We acknowledge this critical human review stage was not sufficiently emphasized in our original manuscript. The combination of LLM assistance and human expertise ensures both efficiency and accuracy in our annotation process. We will add detailed examples and elaborate on this two-stage process in the revised paper.
>
> Best regards,
>
> The Authors

---

> > ### Comment · Reviewer_adFx · 2024-11-26
> >
> > Thank you for addressing the questions raised. I preserve my rating as additional results would be needed to validate the empirical claims of the paper.

---

> ### Author Response · Authors · 2024-11-27
>
> Dear Reviewer adFx,
>
> We sincerely thank you for your thorough and constructive feedback. We appreciate your recognition of our work's empirical strengths and contributions to adversarial robustness. Below we address your specific concerns:
>
> > Comparison with jailbreak defense baselines (e.g. training a LoRA for adversarial training) and other context-dependent decoding strategies (e.g. controlled decoding) should be presented.
>
> We respectfully note that our paper **already includes extensive comparisons with these baselines**. Specifically:
>
> 1. In Table 2 of Section 4.1, we present comprehensive results comparing HiddenGuard against various approaches, including adversarial training.
>
> 2. In our revised version, we have **added controlled decoding comparisons** through the DeCK baseline. As shown in the expanded Table 2, HiddenGuard consistently outperforms DeCK across all attack methods and models. For example, on LLaMA3-8B-Instruct under the GCG attack, HiddenGuard achieves an ASR of 0.9% compared to DeCK's 11.4%.
>
> > The token-redaction training only baseline is lacking (this is motivated in Section 2, but no results are presented)
>
> We thank you for bringing this to our attention. Actually, this comparison is **already included in our ablation studies in Section 4.2**.
>
> | Metrics    | Hidden Guard | Activator MLP | Activator w/o | Router MLP | Router w/o |
> |------------|--------------|---------------|---------------|------------|------------|
> | Precision  | 0.85         | 0.78          | 0.64          | 0.81       | 0.79       |
> | Recall     | 0.87         | 0.75          | 0.67          | 0.85       | 0.76       |
> | F1 Score   | 0.86         | 0.78          | 0.65          | 0.83       | 0.77       |
>
> Specifically:
>
> - Table 4 shows results for "Router w/o" a.k.a. "Without Router (Activator Only)" which represents the token-redaction-only baseline
> - For better clarity, **we further added an expanded version of this existing table and explanations in the Appendix C.3** that more explicitly labels these configurations:
>
> | Configuration                     | Precision | Recall | F1 Score |
> |-----------------------------------|-----------|--------|----------|
> | Full HiddenGuard                    | 0.85      | 0.87   | 0.86     |
> | Without Activator (Router Only)   | 0.79      | 0.76   | 0.77     |
> | Without Router (Activator Only)   | 0.64      | 0.67   | 0.65     |
> | Activator Replaced with MLP       | 0.78      | 0.75   | 0.76     |
> | Router Replaced with MLP          | 0.81      | 0.85   | 0.83     |
>
> The significant performance drop (over 20 percentage points in F1 score) empirically validates our theoretical motivation from Section 2 about the limitations of token-redaction-only approaches.
>
> Lastly, we have carefully incorporated your suggestions by providing more detailed experimental explanations, adding comparisons, and expanding the analysis. We hope these revisions address your concerns comprehensively. Could you kindly reconsider and potentially raise the overall score based on these improvements?
>
> Best regards,
>
> The Authors

---

> ### Author Response · Authors · 2024-12-01
>
> Dear Reviewer adFx,
>
> As we come closer to the end of the discussion period, please let us know if you have any thoughts regarding our above comment addressing your concerns. We thank you very much for your efforts thus far.
>
> Best regards,
>
> The Authors

---

### Author Response · Authors · 2024-11-24
**Unified Response to Reviewers**

**Dear Reviewers,**

We sincerely thank all of you for your thorough reviews, valuable feedback, and dedication to the community's service. Your insights have been immensely helpful in improving our work, and we deeply appreciate your time and effort.

We are especially grateful for the recognition of our paper’s key strengths, which we summarize here. Reviewers have highlighted the effectiveness of our proposed *HiddenGuard* framework in enabling token-level redaction while **preserving utility and informativeness**, surpassing traditional refusal-based methods. The **empirical robustness** of *HiddenGuard*, demonstrated through strong performance on adversarial benchmarks, was another point of agreement. Moreover, the **innovative architecture design**—integrating the *Prism* router with LoRA-based activators—was praised for achieving nuanced moderation with **minimal interference** to the base model’s capabilities. Finally, our **contribution of a token-level annotated dataset** was noted as a valuable resource for the community, supporting research into **context-aware moderation** and fine-grained safety in LLMs.

Building on these strengths, our work aims to address critical challenges in AI safety. By bridging the gap between safety and utility, *HiddenGuard* not only ensures safer LLM outputs but also provides a practical framework for real-world deployment. We see this work as a foundation for future research in AI alignment and context-sensitive moderation.

To address the constructive feedback from reviewers, we have **carefully revised our manuscript**. Specifically, we have **simplified mathematical formulations** in Section 2 to enhance readability and provided **intuitive explanations** of our methodology. The role of the MLP has been **clarified with additional details** in Section 4.2 and Appendix C.3. To help readers better understand the practical impact of our approach, we have included **detailed inference examples** in Appendix D. Furthermore, the **Related Work section** now discusses additional relevant methods suggested by reviewers, including response-modifying and decoding-time techniques. We also conducted a **comprehensive hyperparameter analysis** to clarify the trade-offs in moderation thresholds. Finally, we have elaborated on our **dataset annotation process**, detailing the robust two-stage approach that combines **LLM assistance and human expertise** to ensure accuracy and consistency. As requested by one of the reviewers, **we have added controlled decoding experiments and related analysis**, as detailed in Section 4.1 (Red Teaming) and Table 2. These results demonstrate the consistent superiority of *HiddenGuard* over controlled decoding methods (DeCK) across various attack scenarios and models.

As the **discussion period is nearing its conclusion**, we kindly encourage reviewers to revisit our responses and the revised manuscript. We hope that these revisions address your concerns and further clarify the contributions of our work. If there are any **remaining questions or suggestions**, we would be humbled and eager to address them promptly.

Once again, we are deeply grateful for your constructive feedback and the opportunity to refine our research. Thank you for helping us improve the clarity, completeness, and impact of this work.

Best regards,

The Authors

---

### Author Response · Authors · 2024-12-02
**Kindly Reminder for Reviewers**

Dear Reviewers,

As we approach the end of the discussion period, we would like to extend our sincere gratitude for your thorough and constructive feedback on our work on the *HiddenGuard* framework. We have **carefully addressed all concerns** raised and **made comprehensive revisions to our manuscript**, including enhanced mathematical explanations, clarified methodology sections, additional controlled decoding experiments, and strengthened empirical analysis.

At this stage, we warmly welcome any **additional thoughts or feedback** you may have regarding our responses and revisions. If you find our revisions and responses satisfactory, you may consider **updating your assessment by clicking the "Edit" button next to your official review** to reflect these improvements.

Thank you again for your invaluable guidance in improving this work.

Best regards,
The Authors

---

### Meta-Review · Area_Chair_WHpK · 2024-12-21

**Metareview:**

The authors propose a token-level masking method to redact the parts of LM outputs that are deemed harmful. Doing so provides more fine-grained control over safe LM generation, compared to standard approaches that tend to treat safe generation as a binary problem (i.e., either answer fully or refuse to answer).

Reviewers agreed that the proposed method is an interesting idea for an important problem. However, there were consistent concerns expressed about the clarity of writing, the positioning relative to prior work (to what extent is this the first token-level safe generation methods? etc.), and the overall persuasiveness of the paper. For example, reviewers were unconvinced by the example given in the introduction as well as Figure 1 (where the redactions don't seem to offer any additional benefit compared to the outright refusal, and in fact might just make the output harder for the user to parse). Overall, the authors are proposing a new class of approaches towards safe generation, which is exciting and potentially impactful -- but then the bar is higher to motivate that their approach is better in some way than existing approaches. Reviewers did not feel that the paper met this bar, and therefore I am unable to recommend acceptance. I encourage the authors to take the reviewer feedback into account for a future submission.

**Additional Comments On Reviewer Discussion:**

Other than clarifications, most of the discussion centered around baselines as well as technical questions about the authors' approach. The authors were generally unable to convince the reviewers to change their minds about the weaknesses the reviewers identified.

---

### Decision · Program_Chairs · 2025-01-22

Reject